# Communication Management Processes of Dentists Providing Healthcare for Migrants with Limited Japanese Proficiency

**DOI:** 10.3390/ijerph192214672

**Published:** 2022-11-08

**Authors:** Rintaro Imafuku, Yukiko Nagatani, Masaki Shoji

**Affiliations:** 1Medical Education Development Center, Gifu University, Gifu 501-1194, Japan; 2Department of Dental Hygiene, University of Shizuoka Junior College, Shizuoka 422-8021, Japan; 3Department of Social and Administrative Pharmacy, Osaka Medical and Pharmaceutical University, Takatsuki 569-1094, Japan

**Keywords:** community dentistry, migrant, cross-cultural communication, plain language, social determinants of health, Japan, globalization, qualitative study

## Abstract

Low health literacy results in health inequity are linked with poor adherence to medical care. In the globalized Japanese context, the number of migrants with Japanese as a second language is increasing year after year. Since limited Japanese proficiency may pose a greater health risk, dentists are expected to manage cross-cultural communication and provide dental care to foreign patients. This study explored dentists’ experiences of treating patients with limited Japanese proficiencies. Semi-structured interviews were conducted with 11 community dentists and the qualitative data were analyzed through a thematic analysis approach. Their major challenges were classified into three themes—linguistic aspect (e.g., complicated explanation regarding root canal treatment), sociolinguistic aspect (e.g., communication with foreign residents with limited dental knowledge), and sociocultural aspect (e.g., cultural differences in their dental aesthetics and insurance treatment system). Several management strategies were employed, including linguistic accommodation, avoidance of complexities, use of various communication tools, and getting help from others. However, they were unsatisfied with their practice because they could not understand the patients’ psychosocial aspects due to incomplete communication. These findings provided insights into dentists’ practice in the globalized context.

## 1. Introduction

The social determinants of health are the conditions in which people are born, grow up, work, live, and age and include a broad range of socioeconomic, environmental, political, and cultural factors that influence health status and equity [1]. Globalization, which is an essential aspect in migratory processes, affects disparities in access to the social determinants of health [2]. Cultural and linguistic barriers limit behavioral choices for migrants. Particularly, migrants’ poor proficiencies in the host society’s language hinders them from improving their health literacy and access to health services [3]. Moreover, a greater amount of poor oral health behaviors, including inadequate oral hygiene habits, were reported among the migrants compared to the host population [4,5]. This might be affected by their social and economic factors [6,7]. That is, migrant workers with higher income and education were more likely to use high-quality health services [6]. In a globalized society, looking exclusively at biological factors limits the promotion of individuals’ health and health services are required to consciously address social determinants [8].

Many previous studies investigated migrant patients’ perspectives on patient–provider communication and access to health services [9,10,11]. A study found that Polish migrants in Norway recognized their language competence, lack of knowledge about the health care system, and attitudes among health personnel toward migrants as barriers to the use of healthcare services [9]. Recent studies have increasingly addressed the perspectives of health professionals providing healthcare to migrants and reported the educational development of intercultural communication [12,13,14,15]. In a systematic review by Suphanchaimat et al. [13], practitioners’ attitudes and practices for the provision of healthcare services for migrants were influenced by diverse cultural beliefs and language differences, limited institutional capacity, and the contradiction between professional ethics and laws that limited migrants’ rights to health care. Furthermore, a study by Lindenmeyer et al. [15], exploring the experiences of primary care professionals who provided care for recent migrants, found that practitioners’ motivations and interests, along with working to overcome language barriers through continuous relationships with trusted interpreters and team-based approaches, were identified as facilitators of successful care. However, studies from the perspective of health professionals are still limited. It is necessary to build upon the solid research base by further examining this issue from the health professionals’ perspective to support their practice and educate future professionals in a globalized society.

An effective communicative strategy, in a cross-cultural situation, is using plain language [16,17]. It is the clear and concise language accommodation, for example, by using short sentences and simple words, that mitigates the barriers posed by limited health literacy and enhances patient safety. Warde et al. noted that the aim of this approach was not to oversimplify the health information, but rather to convey essential information in a simple, succinct, and accurate manner [16]. Previous studies reported the development of plain language communication training in medical education [18,19,20]. For example, Bittner et al. provided a newly established course for patient-centered communication, which incorporated the translation practice of medical reports into plain language under near-peer supervision [19]. Their questionnaire survey indicated that students rated themselves significantly higher in most aspect of patient-centered communication after attending this course [19]. 

The majority of research participants in previous studies were physicians and nurses. Although the oral cavity is a mirror of the rest of the body, as Sir William Osler recognized [21], dentists’ communication management processes for migrant patients with limited proficiency in the host society’s language have not been investigated. Similar to other health professionals, dentists are expected to manage cross-cultural communication in oral healthcare and provide treatment and care for migrant patients in a globalized society. Therefore, exploring the actual experiences of migrant patient encounters is pivotal to the development of intercultural communication education in community dentistry and for reducing health inequities. Thus, this study aims to investigate the challenges that dentists face when providing oral treatment and care to migrants with limited proficiency in the host society’s language and how they manage these difficulties in cross-cultural settings.

## 2. Migrants and Healthcare System in Japan

Foreign residents in Japan amounted to 2.03 million in 2012. The number increased to 2.76 million by March 2022, which is about a 2.2% growth in the population (124.75 million) [22]. Chinese migrants are the largest group in Japan, followed by Vietnamese, Korean, Filipino, and Brazilian migrants [22]. Several migrants from these countries are non-native English speakers and use Japanese as a second language in daily life, with varying Japanese proficiency levels [23]. Migrants in Japan, particularly from Asian countries, are mainly international students, technical trainees, spouses of Japanese nationals, restaurant managers, or employees of companies in manufacturing, information technology, or commerce and trade industries [24]. The average monthly salary of migrants in 2021 is 228,100 JPY, whereas that of the Japanese is 307,400 JPY [25]. A survey demonstrated that the migrants recognized issues regarding health and well-being as the most challenging aspect of living in Japan, including healthcare communication and their concerns about health, childbirth, and childcare [26]. In addition to language-related issues, foreign patients’ perceptions of quality healthcare in Japan are influenced by several sociocultural factors, including differences in the medical systems, expectation mismatch between patients and providers, and previous experiences with medical treatments [27,28].

Foreign residents are required to obtain the public national health insurance in Japan. Patients pay 30% of the total cost to the clinic. Dental services under the health insurance system are available for most restorative, prosthetic, and oral surgery treatments, such as fillings, endodontic treatment, crowns, bridges, dentures, and extractions. However, higher cost items (e.g., gold crowns and bridges, metal plate dentures, implants, and orthodontic treatment) are excluded. Preventive services are also excluded, as the current health insurance system only covers treatments for existing diseases [29].

## 3. Materials and Methods

### 3.1. Research Approach

This study adopted an exploratory case study approach, which was informed by an interpretivist paradigm, for an in-depth analysis of the complex phenomenon of migrant patient encounters in community dentistry. Yin defines a case study as “an empirical inquiry that investigates a contemporary phenomenon within its real-life context” [30] (p. 16). The research scope of this qualitative study has been narrowed down to cross-cultural communication between dentists and migrant patients, who are long-term residents, with limited Japanese proficiency in the contexts of dental clinics in Japan.

Language management theory was employed as the conceptual framework. In this theory, comprehensive competence that integrates linguistic, sociolinguistic, and sociocultural aspects is essential for maintaining cross-cultural interactions in contact situations [31,32]. Specifically, the processes of managing problems consist of five stages [31]: (i) deviations from norms occur in a communicative situation, (ii) such deviations are noted, (iii) noted deviations are evaluated, (iv) adjustment is planned, and (v) the adjustment is implemented. All communication problems and phenomena occurring in cross-cultural settings can be related to the above-mentioned stages of the management process.

### 3.2. Participants and Data Collection

This study purposively selected 11 dentists with more than 20 years of clinical experience and who were practicing community dentistry in mid-size regional cities in Japan with some experience in treating and caring for migrants (Table 1). In this study, nine participants were male and two were female dentists, aged from the mid-40s to the early 60s (average age 52.1 years). None of them had any long-term international experience. Their migrant patients primarily came from Brazil, China, Indonesia, Nepal, and Vietnam. According to the research participants, the patients’ occupations are mainly factory workers, restaurant employees, international students, and homemakers.

Semi-structured interviews with the dentists were conducted in person or online, which lasted 30–50 min each. They were asked to share their experiences of cross-cultural communications with migrants with limited Japanese proficiencies, including the difficulties and management strategies used during these encounters. The interview questions were developed based on language management theory to show the processes of managing problems in cross-cultural situations. The main questions are as follows: Please share about the migrant patients with whom you have dealt, such as their countries, ages, the average frequency of visits per week, and their Japanese language proficiency level.What were your most memorable experiences in treating and caring for migrant patients? Please share in detail.If you have been seeing the same patients regularly, how did you build a relationship with these patients?Have you experienced any difficulties or challenges when communicating with migrant patients? Please share in detail.Did you have any special considerations for migrant patients? Please share in detail.

### 3.3. Data Analysis

The interviews were audio-recorded and produced verbatim as transcripts by the researchers. The excerpts of Japanese transcripts were translated into English by the first author. Subsequently, all members discussed the accuracy of the translation and reached a consensus on the final version. Translated excerpts were proofread by a professional English editing company. During this process, private identifiers were replaced with anonymized data, such as D1. For reporting this research in an audit trail, this study maintained careful documentation of all the components of the data analysis process, including raw data, coded transcripts, researchers’ notes, and analysis products.

This study employed Braun and Clark’s reflexive thematic analysis in an inductive way for data analysis [33,34]. Following the six-phase thematic analysis, developed by Braun and Clarke, all the researchers (RI, YN, and MS) systematically reviewed the transcribed data to better understand its content. This is called the familiarization phase. The second phase is coding, where the text data were broken into small units according to their beliefs, actions, events, or ideas. In this phase, RI and YN individually performed initial coding for all the participants. The third phase is generating the initial theme. In this phase, all the members compared the results of individual initial coding and identified significant broader meaning patterns (i.e., theme). Based on this, RI coded the rest of the transcribed data from the 11 dentists. Specifically, each small unit was coded with an interpretive description and was grouped into abstract themes of experiences of migrant patient encounters through the comparison of similarities and differences. The fourth phase is reviewing themes, whereby all the researchers iteratively reviewed the initial themes developed in the previous phases to ensure that the interpretation was congruent with the presented data. The researchers defined the final themes in the fifth phase, which involved developing a detailed analysis, determining the focus, and establishing the story of each theme. Finally, in the sixth phase of writing up, the researchers worked on contextualizing the analysis of the existing literature. The Standards for Reporting Qualitative Research were used for writing the report [35].

This study was approved by Gifu University Institutional Review Board (No.2020-037). Confidentiality was assured for the content of their interview regarding patient–provider communication.

## 4. Results

### 4.1. Overview of Dentists’ Management Process in Intercultural Communication

Figure 1 presents the dentists’ communication management processes during cultural contact. The dentists determine whether the linguistic, sociolinguistic, and sociocultural challenges can be managed. If yes, they try to overcome the challenges by adopting linguistic accommodation, communication tools, and sharing cultural interests to build rapport with the patients. However, if the challenges cannot be managed, they avoid complexities or seek help from others. When the “get-by” strategies do not work, dentists use avoidance of complexities as a management strategy. Nevertheless, in avoiding the complexities, the dentists felt unsatisfied with their practice for the migrant patients. 

The following sections provide the details of dentists’ experiences of providing oral treatment and care for migrant patients.

### 4.2. Challenges during Migrant Patient Encounters

In this study, the dentists did not have any major problems when working on a simple treatment and care for migrant patients, such as early cavities. However, they faced challenges when providing complex oral treatment and care, such as root canals, for migrant patients with limited Japanese proficiencies. This study categorized these challenges into three aspects—linguistic, sociolinguistic, and sociocultural issues. 

#### 4.2.1. Linguistic Issue

The dentists struggled with pronunciation, lexicon, syntax, and listening and speaking during conversations, including dentists’ lack of vocabulary for the patient’s language, difficulty in expressing complex, conceptual phenomena and procedures in simple Japanese, and difficulty in understanding patients’ feelings and symptoms in detail. For example, D7 found it difficult to explain the oral condition to a Brazilian patient and provide instructions for wearing dentures in simple Japanese. Moreover, although D7 recognized the necessity of a long-term treatment plan for the Brazilian patient, he had no way to explain this matter to the patient. In the end, he gave up on discussing the treatment plan and created the “temporal” dentures.


*I made dentures for a Brazilian patient. He had already lost some teeth, and his oral condition was terrible. … As it was the first time for him to wear dentures, he took a long time to do it, … and felt uncomfortable with them. Eventually, he gave up wearing the dentures, and returned home immediately after payment. If you think about his oral condition after 5 years or 10 years, these “temporal” dentures will not work, but I could not explain the long-term vision of his oral condition and treatment plan.*
(D7)

#### 4.2.2. Sociolinguistic Issue

Sociolinguistic issues relate to knowing whom to speak with and when, about what, and how to speak. This partially overlaps with the linguistic and sociocultural aspects as it addresses communication concerning the patients’ status, expectations, and perceptions of the dentists’ roles. For example, the dentists found it difficult to communicate with migrants with limited dental knowledge and inadequate oral hygiene habits. Communicating effectively with such patients to explain the treatment plan and encourage them to adopt better oral health habits is challenging. D6 had difficulty when there was a gap between the patient’s expectations and his treatment plan. Specifically, D6 felt that many migrant patients wanted the decayed tooth to be extracted to relieve toothache, whereas he, as a dentist, wanted to treat these teeth as much as possible.


*When foreign patients have a toothache, they want to pull it out right away because they couldn’t bear the pain, but, as a dentist, I see that some teeth can be treated without extraction. In such cases, it is very difficult to explain and persuade the patient to have dental treatment. Umm, if the patient was Japanese, it would be much easier to convey my opinion. As a dentist, I cannot extract teeth simply because of a toothache.*
(D6)

#### 4.2.3. Sociocultural Issue

Sociocultural issues include difficulties caused by different cultural values, norms, and social systems between Japan and the patients’ home countries. For example, D5 thought that the migrants prefer white filling for their dental aesthetics. However, white fillings for the back tooth are not covered by health insurance in Japan, therefore, they need to pay for this treatment out of their pocket. D5 struggled to deal with this case, complexly intertwining their aesthetics, cultural values, expectations, and health insurance coverage. 


*One of the biggest challenges is explaining health insurance system in Japan to foreign patients. Particularly, people from Brazil dislike silver fillings. … So, if they wish to have white fillings, it would be at their own expense. In this situation, it is difficult to explain why it cannot be offered within the insurance application range.*
(D5)

### 4.3. Communication Management Strategies

Responding to the identified problems, the dentists planned and implemented adjustments for the cross-cultural encounters. This study identified five management strategies that they employed, including linguistic accommodation, the use of various communication tools, sharing interests to build rapport, seeking help from others, and avoiding complexities.

#### 4.3.1. Linguistic Accommodation

Dentists employed linguistical and para-linguistical adjustment strategies to help in cross-cultural encounters, such as speaking slowly, using short phrases and words, mixing Japanese with English vocabulary, written conversation, and expressing it clearly. Moreover, if a patient was from China, the dentists communicated by writing Chinese characters, which are ideographs. 

For example, although the Japanese language, culturally, prefers indirect expressions, the migrants could not fully understand this nuance. Therefore, D11 avoided indirect expression regarding dental treatment and paid attention to straightforwardly explaining tooth extraction. 


*If the patients are Japanese, I can say to them, “I’ll do my best” or “I try to do what I can” because the subtle nuance of my meaning is conveyed to them. I feel that I need to express my opinion clearly and in a straightforward manner to migrant patients …For example, if I say, “I’ll make an effort not to extract your tooth,” the migrants may not fully understand my meaning. So, I would say that “I will pull out your tooth when I find it difficult to treat your tooth.”*
(D11)

#### 4.3.2. Use of Communication Tools 

The dentists adopted several communication tools, such as translation apps, visual information (i.e., illustration, image, and video), and multilingual manuals for dental communication. In case a dentist found it difficult to “verbally” explain the complex treatment procedures and materials, another medium of communication, particularly visual information, was employed. For example, D2 visually explained root canals to migrant patients by using a video of treatment planning. This was useful in explaining the procedure. Nevertheless, if the patients had further questions and opinions, the interactive conversation was difficult to develop.


*We have dental practice management software, which includes visual treatment planning with explanation videos. Using this software, various treatment procedures, such as root canals, and types of fillings can be visually explained. If I really want the patients to understand what I explain, I use this software. … I think it works to some extent.*
(D2)

#### 4.3.3. Sharing Interests

D3 emphasized the importance of sharing common interests to build a better relationship with migrant patients. Particularly, D3 thought that many migrant patients feel nervous when seeing dentists or doctors due to different cultural contexts. A better understanding of the patient’s country and a casual conversation would be important for a relaxed atmosphere. 


*If the patient is from Brazil, I will talk about football in the opening conversation. … I think, a casual conversation allows me to build a better relationship with the patient in a relaxed mood. … The patient may feel nervous in a dental clinic in Japan due to a different language and system. So, soothing their anxiety and making them feel relaxed are key to build rapport.*
(D3)

#### 4.3.4. Seeking Help from Others 

As Figure 1 showed, when dentists found that the challenges could not be managed or the “get-by” strategies did not work, they sought help from others, such as dental staff with international experience, the patient’s family members and friends who can understand Japanese, and professional interpreters. For example, D9 experienced difficulties in communicating with migrant patients with limited Japanese proficiency and requested them to visit the clinic with a person who could be an “interpreter.” D9 said:


*If I can’t communicate on their first visit, I explicitly say, “please come back with someone who can speak Japanese and your language because I am unable to have one-to-one communication with you.” This is the best way. When receiving a booking call from foreign patients, I ask the clinic staff to explain that we request them to come with an interpreter if their Japanese is not very good.*
(D9)

On the other hand, some dentists noted the difficulties in communicating through an interpreter. Specifically, when the interpreter is a patient’s family member or friend, the dentists are unsure about the translation accuracy. D3 said: 


*I’m not sure to what extent the interpreter, like the patient’s family and friends, can understand what I said. Moreover, when the interpreter translates my words into the patient’s language, I totally don’t know the accuracy of the translation. In the end, I’m worried whether the patient understood what I’m saying through an interpreter.*
(D3)

#### 4.3.5. Avoidance of Complexities

Avoidance of complexities includes simplifying the content of the explanation, omitting the details, and providing the most necessary explanation and treatment related to the chief complaint. In most cases, for dentists, the minimum necessary treatment is the last resort and would be performed against their intention because they believe that dentists must improve the patient’s oral health, as a whole, in addition to the chief complaint. For example, D8 provided only the tooth treatment related to the patient complaint even though he identified the necessity of treating other decayed teeth. D8 said:


*Some foreign patients strongly said, “treat only these teeth,” “don’t treat others except this even though you found a decayed tooth in my mouth,” and “just remove the toothache.” For them, I don’t suggest the necessity of treatment for the other parts and treated only their complain. In fact, I should have removed the tartar and treated other decayed teeth. I needed to provide complete oral care to them. However, many foreign patients request me to do only what they want to be treated.*
(D8)

Likewise, D1 selected a short-term treatment plan rather than continuous treatment to improve the patient’s oral health due to communication challenges in a cross-cultural setting. To avoid further problems caused by such communication challenges, D1 had to treat only the tooth directly related to the complaint. D1 said:


*I was not sure how much a Brazilian patient can understand what I explained. So, in this case, I selected not a continuous treatment but a treatment, which could be completed as quickly as possible. For example, I used an inlay and finished the treatment on the next visit. That’s it! So, I just focus on handling the patient’s chief complaint rather than focusing on the whole oral cavity.*
(D1)

When the minimum necessary explanation and treatment were performed, the dentists felt unsatisfied with and regretted their own practice. In fact, they wanted to provide proper treatment to the migrant patients as they provide to the Japanese patients. However, they could not do so due to the challenges of the complex situation. In other words, they were in a dilemma. Regarding this issue, D4 said:


*The foreign patients might want to say something to me. After the dental treatment, I really feel like it was unfinished business. If the patient is Japanese, I can convey what I want to say. … Actually, I want to treat them as I do Japanese patients, but I have no choice but to focus on completing their treatment quickly. That’s why I’m unsatisfied with my practice with the foreign patients.*
(D4)

## 5. Discussion

The present study investigated dentists’ management processes for communicating with migrants with limited Japanese proficiencies. Dentists employed several management strategies to deal with linguistic, sociolinguistic, and sociocultural issues. These findings are congruent with previous studies that explored the experiences of other health professionals in providing treatment and care for migrants [13,14,15]. In other words, the present study has confirmed that dentists, too, adopted similar management strategies as other professionals, including physicians. Moreover, we added new information to the existing knowledge by describing the specific challenges and management strategies employed by the dentists based on the qualitative data of their real experiences and feelings.

Although the dentists in this study accepted migrants in their oral healthcare practice, they faced a dilemma between what they could do practically and what they wanted to do ideally during their practice [14]. The participants did not refer to system-related factors, identified by previous studies, that influenced the reluctance in accepting migrants, such as fear of financial loss, the limited availability of clinicians, and interpreting services [15,36]. Rather, provider-related barriers, such as their own intercultural communication skills, were emphasized. Particularly, the sociolinguistic and sociocultural issues, which derived from different beliefs and values of oral health and healthcare systems, were influential to their practice in cross-cultural settings. This might be related to their limited experience of working or studying overseas. Mota et al. argued that the international experience of health professionals is one of the factors associated with accepting migrants [36]. Although encouraging dentists and dental students in Japan to gain international experience is unrealistic, cultivating a global mindset and cultural competency in undergraduate and clinical education is key to overcoming the challenges of providing treatment and care for migrants [37,38].

In terms of health equality, the minimum necessary treatment provided by the dentists is problematic. To ensure the quality of oral care for migrants and reduce dentists’ psychological burden, the development of medical interpretation services is important [39,40]. There is little compelling evidence that the use of interpretation services has negative impacts on migrant patient visits [14]. However, as opposed to other countries, Japan does not have a legal framework for mandating the provision of interpretation services to patients with limited language proficiencies. Moreover, medical interpreter training may not be able to keep up with the fact that migrants in Japan came from 194 regions and countries [22]. Therefore, interpretation services need to be designed and implemented in the context of wider, political, and institutional developments. Given this context, the dentists felt that the accessibility to the interpretation services is limited and they had to rely on ad hoc interpreters (i.e., the migrant’s family members and friends). The results that certain participants worried about the translation accuracy imply the necessity of further research for examining the interaction patterns in dentistry clinical care [41]. 

It is important to support the dentists to “get by” the migrant patient encounters. Of the three main “get-by” strategies in this study, language accommodation is crucial for providing oral treatment and effective care for migrants, which echoes the findings of previous studies [14,15]. As the national survey demonstrated [23], many migrants can understand simple Japanese in daily life even though their proficiency is limited. In other words, plain language, which is defined as clear and concise language accommodation, can be an effective medium of communication between dentists and migrants [17]. In fact, some dentists in this study use easy words, short phrases, simple sentence structures and clear, assertive expressions in Japanese. For successful language accommodation, plain language needs to be taught systematically in health communication courses in dental education in Japan. There is a growing body of evidence indicating that the incorporation of plain language training into medical education has positive implications for oral and written communication skills, such as breast cancer genetic counseling [20] and discharge letters [42].

The patients’ socioeconomic statuses and educational backgrounds are an important aspect of the quality of treatment and care [6,7]. According to the interview data in this study, most migrant patients treated by the dentists had a blue-collar job with a relatively lower income than local citizens. As Majid et al. [7] argue, ethnic minority migrants in economically developed countries demonstrate higher rates of frailty and are more likely to be frail when younger. Therefore, in addition to sociocultural factors, dentists need to take the socioeconomic background of the migrant patients into account in practice. From a broader perspective, Li et al. emphasized the importance of policymakers’ strengthening health education and increasing medical subsidies to achieve health equality among migrant workers as well as between migrant workers and local citizens [6].

This study is, to best of our knowledge, the first qualitative inquiry focusing on the perspectives of dentists in Japan about their experience of providing oral treatment and care to migrant patients with limited Japanese proficiency. The analysis of dentists’ reflections on their cross-cultural experiences allowed the researchers to examine the processes of their communication management in community dentistry. However, there are certain limitations to the study. The results were not generalizable due to the small number of participants from dental clinics in a regional city in Japan. Particularly, the different contexts, such as metropolitan cities and rural areas, may lead to different results due to differences in human resources, availability of communication tools and interpreter services, and demographics of the migrant population. Thus, based on the findings of this study, further surveys across Japan need to be designed and conducted regarding the status quo of providing oral treatment and care for migrant patients. At the same time, fine-grain analyses need to be continuously conducted to include actual dentist experiences, such as social interactions during migrant patient encounters. Furthermore, as the data were collected only from the dentists, the perspectives of migrant patients and other dental staff need to be further explored.

## 6. Conclusions

This study revealed the process by which the dentists managed to provide oral treatment and care for migrant patients with limited Japanese proficiency. The dentists encountered linguistic, sociolinguistic, and sociocultural challenges and tried to overcome them by employing several strategies. When the minimum necessary explanation and treatment were provided, the dentists felt unsatisfied with their practice and faced the dilemma between real and ideal practice. The findings of this study have provided new insights into dentists’ practices in the globalized context. Particularly, the use of plain language, as a part of language accommodation, has the potential to enhance dentists’ cross-cultural communication with migrant patients. 

## Figures and Tables

**Figure 1 ijerph-19-14672-f001:**
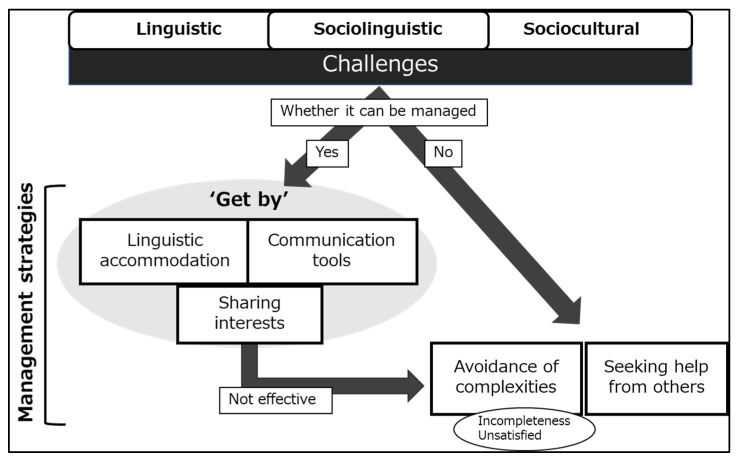
Overview of dentists’ communication management process.

**Table 1 ijerph-19-14672-t001:** Research participants.

	Gender	Clinical Experience	Patients’ Countries
D1	Male	26 years	Brazil, China, Vietnam, United States
D2	Female	22 years	Vietnam
D3	Male	24 years	Brazil, Vietnam
D4	Male	38 years	Brazil, China, Mauritius
D5	Male	20 years	China, Sri Lanka, United States
D6	Male	34 years	Brazil, China, Vietnam
D7	Male	33 years	Brazil, China, Indonesia, Italy, United States
D8	Female	32 years	Brazil, Indonesia, Thailand, Vietnam
D9	Male	26 years	Brazil, China, Nepal, Russia, United States,
D10	Male	30 years	Brazil, United States
D11	Male	23 years	Brazil, China, Indonesia, Peru

## Data Availability

The data presented in this study are available on request from the corresponding author.

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
