# Peer review of "(untitled)"

_ijerph, 2022, doi:10.3390/ijerph192214672_

Round 1

Reviewer 1 Report

The manuscript deals with the problems of communication with immigrant patients in the clinical practice of oral health in Japan, and the strategies to face it, according to the professionals' point of view. The methodological approach is appropriate to get to know in depth the difficulties that dentists encounter with this type of patients and to reveal the processes that lead to inequalities in the quality of care in this population.

However, there are some aspects that can be improved:

Introduction:

The authors should explain in more detail how the health system works to provide oral health coverage to the immigrant population.

They should also justify why they focus on the professionals' point of view in this study, and do not include the patient's perspective, although they include it as a limitation in the discussion section.

There is a lack of greater focus on the socioeconomic status and educational level of immigrant patients.

Line 31-32: It would be necessary to debate whether Globalization is the key to the social determinants of health. Rather, it could be said that it is an essential aspect in migratory processes.

Line 32: please, delete "For example" to highligh the research topic.

Line 37: "In other words" doesn't seem like an expression for the meaning of the sentence. I think it should also be deleted for better understanding.

Line 37: What do the authors mean by "health anxiety"?

Line 77: Consider changing "mouth" to "oral health".

Methods:

Lines 102-103: Please, detail the range and average of age in years of the healthcare givers.

Line 128: Who did the transcription? Please explain how anonymity was managed in this phase of the analysis.

Results:

Line 296: Correct the typo of "requwsted".

discussion:

The work has a culturalist bias that overshadows the role of social class. It should include a reflection on the role of the socioeconomic status and the educational level of the patients treated by the dentists interviewed in relation to the differences in the quality of care provided that the professionals recognize.

Reviewer 2 Report

This is an interesting article. Prior to consideration for publication, the following points should be addressed:

1) Please provide a citation to justify the following statement in the introduction - "Globalization, including population mobility, is the key aspect of social determinants of health."  

2) The introduction is perhaps a bit to lengthy. A condensed version would benefit the readers.

3) Further description as to how the semi-structured interview questions were developped is needed in the methods section

4) Was there any process to verify the translation of the transcripts? Further details are required as this is a critical element.

5) The results section is needless long. In some cases, the quotations are longer then the descriptive paragraph. While the use of direct quotations is an important part of qualitative research, their use here is perhaps superfluous. I suggest trying to condense the overall results section so that only the most important points are highlighted.

Round 2

Reviewer 2 Report

The manuscript is improved after the revisions.